# K^+^-Driven Cl^−^/HCO_3_^−^ Exchange Mediated by Slc4a8 and Slc4a10

**DOI:** 10.3390/ijms25084575

**Published:** 2024-04-22

**Authors:** Gaspar Peña-Münzenmayer, Alvin T. George, Nuria Llontop, Yuliet Mazola, Natalia Apablaza, Carlos Spichiger, Sebastián Brauchi, José Sarmiento, Leandro Zúñiga, Wendy González, Marcelo A. Catalán

**Affiliations:** 1Instituto de Bioquímica y Microbiología, Facultad de Ciencias, Universidad Austral de Chile, Valdivia 5090000, Chile; gaspar.pena@uach.cl (G.P.-M.); natalia.apablaza11@gmail.com (N.A.); cspichiger@uach.cl (C.S.); 2Millennium Nucleus of Ion Channels-Associated Diseases (MiNICAD), Valdivia 5090000, Chile; 3Secretory Mechanisms and Dysfunction Section, National Institute of Dental and Craniofacial Research, National Institutes of Health, Bethesda, MD 20892, USA; 4Instituto de Fisiología, Facultad de Medicina, Universidad Austral de Chile, Valdivia 5090000, Chile; nllontoplopez@gmail.com (N.L.); jsarmien@uach.cl (J.S.); 5Center for Bioinformatics and Molecular Simulations (CBSM), Universidad de Talca, Talca 3460000, Chilewgonzalez@utalca.cl (W.G.); 6Centro de Nanomedicina, Diagnóstico y Desarrollo de Fármacos (ND3), Laboratorio de Fisiología Molecular, Escuela de Medicina, Universidad de Talca, Casilla, Talca 3460000, Chile; lzuniga@utalca.cl; 7Millennium Nucleus of Ion Channels-Associated Diseases (MiNICAD), Talca 3460000, Chile

**Keywords:** K^+^-driven anion exchanger, chloride/bicarbonate exchangers, sodium/bicarbonate cotransporters, ion transport

## Abstract

Slc4a genes encode various types of transporters, including Na^+^-HCO_3_^−^ cotransporters, Cl^−^/HCO_3_^−^ exchangers, or Na^+^-driven Cl^−^/HCO_3_^−^ exchangers. Previous research has revealed that Slc4a9 (Ae4) functions as a Cl^−^/HCO_3_^−^ exchanger, which can be driven by either Na^+^ or K^+^, prompting investigation into whether other Slc4a members facilitate cation-dependent anion transport. In the present study, we show that either Na^+^ or K^+^ drive Cl^−^/HCO_3_^−^ exchanger activity in cells overexpressing Slc4a8 or Slc4a10. Further characterization of cation-driven Cl^−^/HCO_3_^−^ exchange demonstrated that Slc4a8 and Slc4a10 also mediate Cl^−^ and HCO_3_^−^-dependent K^+^ transport. Full-atom molecular dynamics simulation on the recently solved structure of Slc4a8 supports the coordination of K^+^ at the Na^+^ binding site in S1. Sequence analysis shows that the critical residues coordinating monovalent cations are conserved among mouse Slc4a8 and Slc4a10 proteins. Together, our results suggest that Slc4a8 and Slc4a10 might transport K^+^ in the same direction as HCO_3_^−^ ions in a similar fashion to that described for Na^+^ transport in the rat Slc4a8 structure.

## 1. Introduction

The regulation of pH is critical for physiological processes such as synaptic activity and transepithelial transport. Carbonic anhydrases and H^+^-transporting proteins (i.e., Na^+^/H^+^ exchangers and H^+^-ATPases) are key proteins involved in pH homeostasis, which often are functionally coupled with other ion-transporting proteins such as Na^+^-HCO_3_^−^ cotransporters, Cl^−^/HCO_3_^−^ exchangers and Na^+^-dependent Cl^−^/HCO_3_^−^ exchangers [1,2].

With the sole exception of Slc4a11, which does not transport HCO_3_^−^ [3,4,5,6], all the other members of the *Slc4a* gene family encode for HCO_3_^−^ transporters with different activities (i.e., Na^+^-HCO_3_^−^ cotransporters, Na^+^-independent and Na^+^-dependent Cl^−^/HCO_3_^−^ exchangers) [7]. The importance of Slc4a transporters in pH homeostasis can be inferred from pathologies associated with *SLC4A1-5*, *SLC4A7* and *SLC4A10* gene mutations [8,9].

Due to the large Na^+^-inward electrochemical gradient displayed by the majority of mammalian cells, electroneutral Na^+^-driven Cl^−^/HCO_3_^−^ exchangers are believed to mediate HCO_3_^−^ influx under physiological conditions. However, Na^+^-driven inward HCO_3_^−^ transport might not be a common theme for all Slc4a members. For instance, intracellular [Cl^−^] at the resting state is reduced in acinar secretory cells from *Slc4a9^−/−^* (*Ae4*) salivary glands, suggesting that Slc4a9 promotes Cl^−^ influx rather than HCO_3_^−^ influx [10]. A more recent study showed that Slc4a9 is in fact a cation-driven Cl^−^/HCO_3_^−^ exchanger displaying a cation selectivity, Na^+^ = K^+^ [11]. Therefore, it is likely that other Slc4a members that mediate Na^+^-driven Cl^−^/HCO_3_^−^ exchange might also transport K^+^ ions.

In the present study, we evaluated the cation selectivity of the Na^+^-dependent transporters Slc4a8 and Slc4a10 by using a functional fluorescence approach. We found that both Slc4a8 and Slc4a10 display significant Cl^−^/HCO_3_^−^ exchanger activity in the presence of either Na^+^ or K^+^ ions. Molecular dynamics simulations, based on the available Slc4a8 structure confirm that Slc4a8 has the ability to coordinate K^+^ ions in a similar way to how Na^+^ ions are coordinated within the transporter’s structure.

## 2. Results

### 2.1. Expression of Mouse Slc4a8 and Slc4a10 in CHO-K1 Cells

We first evaluated by indirect immunofluorescence the expression of the Myc-DDK tagged versions of mouse Slc4a8 and Slc4a10 clones in CHO-K1 cells. Slc4a8- and Slc4a10-tagged proteins were localized as punctate signals within the intracellular space as well as in the cell periphery (Figure 1, green fluorescence).

To visualize the plasma membrane, transfected CHO-K1 cells were stained with wheat germ agglutinin (WGA) conjugated to Alexa Fluor^TM^ 633 (red fluorescence). Some overlap between WGA- and Slc4a8- (Figure 1, upper panels) or Slc4a10- (Figure 1, middle panels) associated fluorescence was observed, confirming the membrane expression of the recombinant proteins.

The specificity of the anti-DDK antibody was verified by the absence of immunoreactivity in CHO-K1 cells transfected with the empty vector (Figure 1, lower panels).

### 2.2. Cl^−^ Fluxes Are Associated with Slc4a8 and Slc4a10 Expression in CHO-K1 Cells

There is no consensus regarding the activity encoded by Slc4a8 and Slc4a10; some reports describe Slc4a8 and Slc4a10 as Na^+^-HCO_3_^−^ cotransporters, while other studies show that Slc4a8 and Slc4a10 are Na^+^-driven Cl^−^/HCO_3_^−^ exchangers [12,13,14,15]. To assess whether Cl^−^ fluxes are associated with the activities of Slc4a8 and Slc4a10, we quantified changes in fluorescence intensity in cells loaded with the Cl^−^ indicator SPQ. As seen in Figure 2A,B, a loss in normalized fluorescence (black circles) occurred in response to a reduction in the external Cl^−^ concentration in Slc4a8- and Slc4a10-expressing cells (~20%) that was substantially higher than that observed in non-transfected CHO-K1 cells (~5%, dashed line). We interpret the loss in fluorescence intensity as an increase in the rate of Cl^−^ efflux from the cells. Moreover, Slc4a8- and Slc4a10-mediated Cl^−^ efflux was dependent on HCO_3_^−^ as little, if any, Cl^−^ efflux was seen in Slc4a8- and Slc4a10-expressing cells measured under HCO_3_^−^-free conditions (open circles). The reduction in normalized fluorescence intensity and the change in flow rates was comparable to that obtained from control cells measured under HCO_3_^−^-containing solutions (dashed lines).

### 2.3. Na^+^- and K^+^-Dependence of Slc4a8- and Slc4a10-Mediated Cl^−^/HCO_3_^−^-Exchange in CHO-K1 Cells

We next evaluated whether Slc4a8 and Slc4a10 activities display a cation dependence. To assess whether HCO_3_^−^ fluxes are associated with the activities of Slc4a8 and Slc4a10, we quantified changes in fluorescence intensity in cells loaded with the pH^−^ indicator BCECF-AM. Figure 3 shows that no alkalinization was observed in response to external Cl^−^ reduction in Slc4a8- and Slc4a10-expressing cells when N-Methyl-D-Glucamine (NMDG) was the main cation present in the external solution. In contrast, a robust alkalinization was evident when external NMDG was switched to either Na^+^ or K^+^ ions in the low Cl^−^ solution. Alkalinization induced by Na^+^ or K^+^ in cells expressing Slc4a8 and Slc4a10 occurs in a concentration-dependent fashion (Figure 3B,D). Hill function analyses of dose–response experiments revealed EC_50_ values of 38.5 mM (Na^+^) and 40.8 mM (K^+^) for Slc4a8, while for Slc4a10, values of 54.5 mM (Na^+^) and 58.0 mM (K^+^) were obtained.

### 2.4. Slc4a8- and Slc4a10-Mediated K^+^ Transport in CHO-K1 Cells

To directly assess whether potassium ions (K^+^) are being transported by Slc4a8 and Slc4a10, we employed the K^+^-selective probe PBFI-AM to measure intracellular potassium concentrations ([K^+^]_i_) during stimulated exchanger activity. Due to the slight preference for K^+^ over Na^+^ ions (~1.5 times) displayed by PBFI, the experiments were conducted under Na^+^-free conditions to eliminate potential interference of the K^+^ signal by Na^+^ ions, as outlined by Minta and Tsien in 1989 [16]. The experimental design, depicted in Figure 4A, is supported by our previous work [11]. In brief, cells were loaded with PBFI-AM and exposed to a low-chloride ([Cl^−^]), Na^+^-free, and high-potassium ([K^+^]) external solution to deplete cellular chloride (Figure 4A, condition i). If Slc4a8 and Slc4a10 facilitate K^+^ transport, it would be anticipated that the combination of inwardly directed Cl^−^ and outwardly directed K^+^ gradients (achieved by increasing [Cl^−^]_o_ and decreasing [K^+^]_o_) would induce Cl^−^ influx in exchange for K^+^ − HCO_3_^−^ efflux (Figure 4A, condition ii). As shown in Figure 4B,C, a sustained reduction in [K^+^]_i_ was observed when these gradients were imposed on Slc4a8- and Slc4a10-expressing cells. In contrast, minimal or no K^+^ efflux was detected in non-transfected cells, strongly supporting the proposition that Slc4a8 and Slc4a10 indeed facilitate the transport of K^+^ ions.

### 2.5. Molecular Dynamics Supports K^+^ Occupancy at the Na^+^ Coordination S1 Site in the Cryo-EM rSlc4a8 Structure

In this study, we have shown evidence supporting K^+^ (or Na^+^)-driven Cl^−^/HCO_3_^−^ exchanger activity by the murine Slc4a8 (mSlc4a8) and Slc4a10 (mSlc4a10) transporters. In the absence of structural data for murine orthologs of Slc4a8 or Slc4a10 proteins, we turned to the accessible cryo-EM structure of Slc4a8 from *Rattus norvegicus* (PDB: 7rtm), recently solved in its outward-facing (OF) state [17].

In the rat Slc4a8 structure (rSlc4a8), Na^+^ and CO_3_^2−^ ions are anchored in the S1 site, which is located at the center of the protein. The Na^+^ position is coordinated mainly by residues D800, T804, T847, and the CO_3_^2−^ anion [17]. The rSlc4a8 shares 97% sequence identity with mSlc4a8 and 74% with mSlc4a10. In addition, the residues D800, T804, and T847 in rSlc4a8 as well as others from the S1 site are conserved in mSlc4a8 (D798, T802 and T845, Appendix A) and mSlc4a10 (D831, T835, and T878, Appendix A).

Using molecular dynamics (MD) simulations, we explore whether K^+^ could also occupy the natural coordination S1 site for Na^+^ in the cryo-EM rSlc4a8 structure. We replaced Na^+^ by K^+^ in the presence of CO_3_^2−^. As result, we obtained a rSlc4a8 structure coordinating the K^+^-CO_3_^2−^ pair, which was used as input for three independent 100 ns MD simulations. In the stable portion of the three trajectories (last 50 ns in Appendix A), the distances between K^+^ and Na-coordinating residues (D800, T804, and T847) were computed to explore the putative coordination of K^+^ ion at the S1 site.

We found that K^+^ remains in contact with D800, T804, and T847 along all MD simulations (Figure 5) and that the K^+^-residue distances (Figure 6C) are comparable to those reported for Na^+^ (Figure 6D) [17]. In addition, we evidenced that K^+^ and CO_3_^2−^ are interacting with an average distance of 3.7 Å (Figure 6C) and that they sustain interactions (Figure 7A) that do not dissociate from the S1 site during 50 ns MD simulations (Figure 7B,C). Distances between each ion to S1 site were calculated from the Cα atom of residue A846 (Appendix A).

In addition, we measured the ion-S1 distances (K-S1 and Na-S1), which correspond to the distances between one permanent ion (Na^+^ or K^+^) and the residue A846, providing evidence of whether ions dissociate from their pocket. The A846 residue belongs to S1 site, and the ion-S1 distances are computed between the ion center of mass and the Cα atom of residue A846 (Figure 7B,C), similar to the definition used by Wang et al. [17].

## 3. Discussion

Slc4a8 expression is associated with Na^+^-driven Cl^−^/HCO_3_^−^ exchanger activity [14]. Additionally, apart from its role as a Na^+^-HCO_3_^−^ symporter with Cl^−^ self-exchange activity [15], Slc4a10 has been shown to exhibit sodium-driven HCO_3_^−^/Cl^−^ exchange activity [13,18]. In summary, the literature suggests that Slc4a8 and Slc4a10 could be involved in regulating intracellular pH and chloride concentrations, thereby implying their contribution to the modulation of synaptic transmission through ion homeostasis mechanisms.

Synaptic glutamate release critically depends on the activity of the Na^+^-driven Cl⁻/HCO₃⁻ exchanger Slc4a8 [19]. Moreover, the absence of Slc4a10 in knockout mice disrupts GABAergic transmission, with glutamatergic transmission unaffected [20]. The importance of Slc4a10 in synaptic function is underscored by revealing impaired acid extrusion in humans with mutations in the SLC4A10 gene [21]. Together, Slc4a8 and Slc4a10 influence neurotransmission at the presynaptic level, where they modulate neurotransmitter release through a mechanism that likely depends on sodium-coupled bicarbonate transport.

Our findings demonstrate that the activities associated with Slc4a8 and Slc4a10 involve HCO_3_^−^-dependent Cl^−^ flux (Figure 2). Furthermore, a decrease in external Cl^−^ concentration led to an elevation in intracellular pH in a HCO_3_^−^-dependent manner in cells expressing Slc4a8 and Slc4a10, indicating Cl^−^/HCO_3_^−^ exchanger activity (Figure 3). Intriguingly, both Na^+^ and K^+^ ions were found to drive Cl^−^/HCO_3_^−^ exchange in cells expressing Slc4a8 and Slc4a10, suggesting that either Na^+^ or K^+^ may be transported in the same direction as HCO_3_^−^ ions (Figure 3). Additional experiments conducted in cells expressing Slc4a8 and Slc4a10 and loaded with the K^+^ indicator PBFI-AM revealed K^+^ fluxes associated with the expression of Slc4a8 and Slc4a10 (Figure 4), indicating that K^+^ ions might be transported by a mechanism similar to that shown for Na^+^ ions transported by the rat Slc4a8 protein [17].

Utilizing the rat Slc4a8 structure [17], we conducted MD calculations, which provide evidence supporting K^+^ binding to the S1 site in rSlc4a8, indicating that favorable interactions for Na^+^ coordination are also favorable to K^+^ binding. Given the conservation of residues in mSlc4a8, mSlc4a10 and rSlc4a8, we anticipate that mSlc4a8 and mSlc4a10 could also bind K^+^ in the sodium coordination pocket using residues D798, T802, and T845 in mSlc4a8, and residues D831, T835, and T878 in mSlc4a10 (equivalent counterparts of D800, T804, and T847 in rSlc4a8, respectively). Moreover, our calculations showed that there is an interaction between K^+^ and CO_3_^2−^ at the S1 site (3.7 Å distance). Collectively, molecular dynamics calculations support the notion of K^+^ occupying a cation transport pathway, thereby mediating Cl^−^/K^+^ -HCO_3_^−^ (or CO_3_^2−^) exchanger activity of mSlc4a8 and mSlc4a10 transporter proteins.

This K^+^ dependence exhibited by Slc4a8 and Slc4a10 is reminiscent of that described for Ae4, a Cl^−^/HCO_3_^−^ exchanger driven by either Na^+^ or K^+^ [11]. However, it is important to note that this non-selective cation feature is not conserved in all Na^+^-driven Cl^−^/HCO_3_^−^ exchangers, as evidenced by the lack of K^+^ driving anion exchange in the squid Na^+^-driven Cl^−^/HCO_3_^−^ exchanger (sqNDCBE) [22].

## 4. Materials and Methods

Plasmids. Empty pCMV6-entry plasmid and pCMV6-entry plasmid encoding *Mus musculus* Slc4a8 Myc-DDK tagged (GenBank Accession no. NM_021530) and Slc4a10 Myc-DDK tagged (transcript variant 5, GenBank Accession no. NM_001242381) were obtained from Origene Technologies Inc. (Rockville, MD 20850, USA).

Cell culture and transfections. CHO-K1 cells (Sigma-Aldrich, Saint Louis, MO 63103, USA) were grown as previously described [11]. Cells were electroporated (Nucleofector II, Amaxa, Gaithersburg, MD 20877, USA) with the plasmids mentioned above (4 µg DNA per reaction) using the Nucleofector Kit V (Lonza, Morristown, NJ 07960, USA) following the manufacturer’s instructions and then seeded onto 5 mm diameter glass coverslips (Warner Instruments, Holliston, MA 01746, USA). The expression of Slc4a8 and Slc4a10 in CHO-K1 cells resulted in comparable yields: 55.7 ± 4.7% for Slc4a8 and 53.4 ± 4.9% for Slc4a10.

For the immunofluorescence experiments (see methods below), CHO-K1 cells were transfected with 2.5 μg of Slc4a8, Slc4a10 or empty plasmids using the Xfect^TM^ (Takara, San Jose, CA 95131, USA) transfection reagent according to the manufacturer’s instructions.

Immunofluorescence experiments. CHO-K1 cells were trypsinized after transfections and seeded onto 25 mm diameter glass coverslips (Superior Marienfeld, 97922 Lauda-Königshofen, Germany). At 24 h after transfection, the cells were incubated for 60 min on ice in complete culture medium supplemented with wheat germ agglutinin (WGA) conjugated to Alexa Fluor^TM^ 633 fluorophore (1 μg/mL, Thermo Fisher Scientific, Waltham, MA 02451, USA), followed by cell fixation with 4% paraformaldehyde (PFA) for 20 min and methanol for 45 min at −20 °C. The coverslips were air dried and then stored overnight at −80 °C. The next day, cells were blocked with 3% bovine serum albumin (BSA) diluted in phosphate-buffered saline (PBS) containing (in mM): 137 NaCl, 2.7 KCl, 8 Na_2_HPO_4_, 2 KH_2_PO_4_, at pH 7.4. After blocking, the coverslips were incubated for 1 h at room temperature (RT) with a mouse monoclonal anti-DYKDDDDK antibody (9A3, 1:500 dilution, Cell Signaling Technologies, Danvers, MA 01923, USA) followed by three 5 min washings with PBS. Then, the coverslips were incubated for 1 h at RT with goat anti-mouse IgG (H + L) conjugated to Alexa Fluor^TM^ 488 (1:1000 Jackson Immunoresearch, West Grove, PA 19390, USA) followed by three washings with PBS. Finally, nuclei were stained with 4’,6-diamidino-2-phenylindole (DAPI 1:50,000 dilution, Thermo Fisher Scientific) and mounted using fluorescence mounting medium (Dako, Santa Clara, CA 95051, USA).

Images were acquired using a DMi8 Leica spinning disk confocal microscope equipped with a HC PL APO 63x×oil immersion objective (N.A.1.40) and an EMCCD digital camera (iXon Life 888, Andor Technology, Belfast, United Kingdom). Images were captured using SlideBook 6 software (Intelligent Imaging Innovations, Denver, CO 80216, USA) and processed with ImageJ software (version 1.54f) [23].

Intracellular pH, [Cl^−^] and [K^+^] measurements. The solutions used in this study are shown in Table 1 and Table 2. Functional experiments were conducted 18–20 h after electroporation. The fluorescent ion-sensitive dyes BCECF-AM (2′,7′-Bis-(2-Carboxyethyl)-5-(and-6)-Carboxyfluorescein, Acetoxymethyl Ester) [24], PBFI-AM (1,3-benzenedicarboxylic acid, 4,4′-[1,4,10,13-tetraoxa-7,16-diazacyclooctadecane-7,16-diylbis(5-methoxy-6,2-benzofurandiyl)]bis-, tetrakis[(acetyloxy)methyl] ester) [16] and SPQ (6-methoxy-N-[3-sulfopropyl] quinolinium) [25] were purchased from Thermo Fisher Scientific and used for measuring intracellular pH, [K^+^] and [Cl^−^], respectively.

For intracellular pH measurements, CHO-K1 cells were loaded with 2 μM BCECF-AM for 15 min at 37 °C (95% O_2_/5% CO_2_). For intracellular K^+^ measurements, CHO-K1 cells were loaded with 10 μM PBFI-AM for 80 min at room temperature (95% O_2_/5% CO_2_). For intracellular [Cl^−^] measurements, CHO-K1 cells were exposed to a hypotonic loading solution containing 10 mM SPQ as previously described [26]. Solutions containing HCO_3_^−^ were gassed with 95% O_2_/5% CO_2_. For experiments performed under HCO_3_^−^-free conditions, solutions were gassed continuously with 100% O_2_ and supplemented with 20 μM ethoxyzolamide (EZA), a carbonic anhydrase inhibitor.

Imaging experiments were performed using an inverted microscope (Olympus IX71, Olympus America Inc., Center Valley, PA 18034, USA) equipped with a Polychrome IV Imaging System coupled to a high-speed digital camera (Till Photonics, Victor, NY 14564, USA). Images were acquired by alternate excitation at 490 and 440 nm (BCECF), 340 and 380 nm (PBFI) or excitation at 340 nm (SPQ). Emissions were captured at 530 nm (BCECF) or 510 nm (PBFI and SPQ) using Imaging WorkBench 6.0 software (INDEC BioSystems, Los Altos, CA 94022, USA) or Micromanager 1.4 software [27]. The temperature of the solutions (listed in Table 1 and Table 2) was kept at 37 °C during the experiments using a CL-100 bipolar temperature controller (Warner Instruments).

Slc4a8 and Slc4a10 dependence on Na^+^ and K^+^ concentrations. The dependence of Slc4a8 and Slc4a10 for Na^+^ and K^+^ ions was obtained by fitting the normalized Slc4a8 and Slc4a10 activities (flow rates correspond to the slopes obtained by linear regression analysis of the linear response recorded when transitioning from an NMDG-containing external solution to either a Na^+^ or K^+^-containing external solution) measured at different [Na^+^]_o_ and [K^+^]_o_ to the following Hill function using Origin 8.0 software (OriginLab):$$R_{Cation}=R_{Min}+\frac{\left(R_{Max}-R_{Min}\right)\times {\left[Cation\right]}^{nH}}{{{EC}_{50}}^{nH}{+\left[Cation\right]}^{nH}}$$
where R_Cation_ is the alkalinization rate in response to a reduction in external [Cl^−^] at different [Cation], and R_Min_ and R_Max_ correspond to the minimal and maximal alkalinization rates. EC_50_ is the concentration required to achieve 50% of the maximal effect and nH is the Hill number.

Molecular dynamics simulations. The cryo-EM structure of the rat Slc4a8 (PDB: 7rtm) with K^+^ in place of Na^+^ at the ion coordination pocket was used as input for molecular dynamics (MD) simulations. A total of three MD simulations were performed for 100 ns each. They were executed using Desmond v2019-1 [28] and OPLS2005 force field [29,30,31]. The Slc4a8 structure with the K^+^-CO_3_^2−^ pair at the S1 site was prepared using Protein Preparation Wizard from Maestro suite [32]. The system was embedded into a pre-equilibrated POPC (1-palmitoyl-2-oleoyl-sn-glycero-3-phosphocholine) bilayer membrane model and solvated using the single point charge (SPC) water model. To neutralize the system, Na^+^/Cl^−^ ions were included; NaCl was added at a concentration of 0.15 M. The system was equilibrated for 20 ns in the NPT ensemble. Positional restraints of 1.0 kcal×mol^−1^×Å^−2^ were applied to the protein and to the ions placed in coordination sites (K^+^ and CO_3_^2−^). Temperature and pressure were kept constant at 300 K and 1.01325 bar, respectively, by coupling to a Nose–Hoover Chain thermostat [33] and Martyna–Tobias–Klein barostat [34]. The integration step was set to 2 fs. For MD productive runs, the positional restraints were removed. Each run was performed during 100 ns using an NPγT (semi-isotropic ensemble) with a constant surface tension of 0.0 bar Å. For each trajectory, the root-mean-square deviation (RMSD) values of the backbone atoms were computed using VMD v1.9.4a38 [35]. The distances between ions and residues were computed using TCL scripting in VMD v1.9.4a38 [35].

Sequence alignment. Multiple sequence alignment was performed using Clustal Omega [36]. Sequences were retrieved from the UniProtKB database and correspond to mouse Slc4a8 (code: Q8JZR6), rat Slc4a8 (code: F1LUB7) and mouse Slc4a10 (code: Q5DTL9).

Statistical analysis. Results are presented as the mean ± standard error of mean (SEM), where n corresponds to the number of experiments per condition. Data were obtained from at least three independent electroporations (or transfections for immunolocalization studies) per condition. Statistical significance was determined using Student’s *t* test and one-way ANOVA analysis followed by Bonferroni’s post hoc test. *p* values of less than 0.05 were considered statistically significant. Origin 8.0 Software was used for statistical calculations (OriginLab, Northampton, MA, USA).

## Figures and Tables

**Figure 1 ijms-25-04575-f001:**
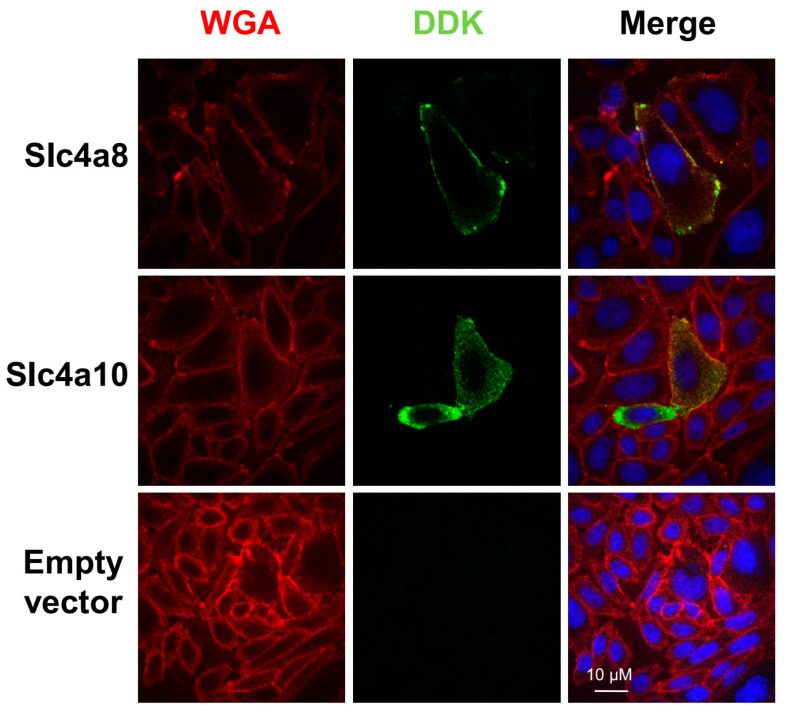
Slc4a8 and Slc4a10 localization in CHO-K1 cells. Immunofluorescence studies in CHO-K1 cells transfected with plasmids encoding Myc-DDK tagged versions of Slc4a8 (upper panels) and Slc4a10 (middle panels). Transfected cells with the empty plasmid were used as negative control (lower panels). Plasma membrane was labeled with WGA conjugated to Alexa Fluor 633^TM^ (shown in red, left panels), followed by protein detection using an anti-DYKDDDDK antibody (shown in green, middle panels). Pictures in the right panels were generated by merging WGA, DYKDDDDK and DAPI channels. Scale bar = 10 μm.

**Figure 2 ijms-25-04575-f002:**
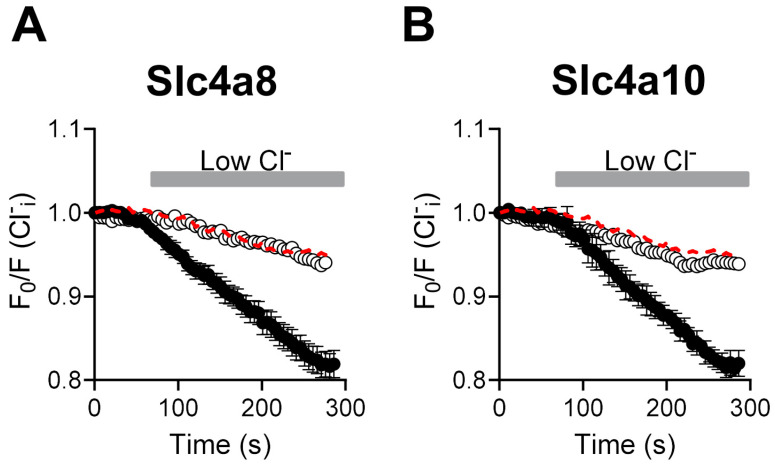
Slc4a8 and Slc4a10 mediate Cl^−^/HCO_3_^−^ exchanger activity. Cl^−^ fluxes in Slc4a8- and Slc4a10-expressing cells loaded with SPQ. Electroporated CHO-K1 cells were perfused with HCO_3_^−^-containing/high Cl^−^ solution (Solution A, Table 1) and then switched to a HCO_3_^−^-containing/low Cl^−^ solution (Solution B, Table 1). Fluxes obtained from cells electroporated with Slc4a8- ((**A**), *n* = 9) or Slc4a10 ((**B**), *n* = 9) encoding plasmids are shown in black circles. Fluxes obtained from non-electroporated cells are shown as red dashed lines and are the same in A and B (*n* = 11). Experiments under HCO_3_^−^-free conditions were also performed on Slc4a8 ((**A**), open circles; *n* = 5) and Slc4a10 ((**B**), open circles; *n* = 5) electroporated cells, which were initially perfused with HCO_3_^−^-free/high Cl^−^ solution (Solution E) and then switched to HCO_3_^−^-free/low Cl^−^ solution (Solution F). Fluxes in each condition are shown as the mean ± standard error of mean (SEM).

**Figure 3 ijms-25-04575-f003:**
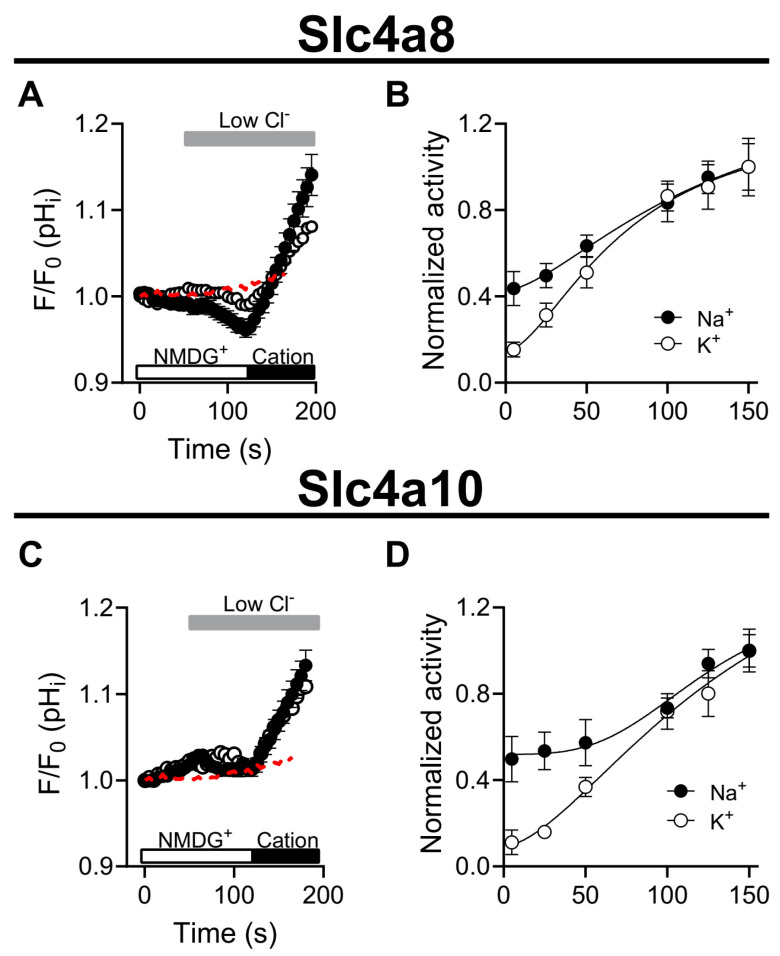
Slc4a8 and Slc4a10 display Na^+^- and K^+^- dependent Cl^−^/HCO_3_^−^ exchanger activity. (**A**,**C**) Time course showing that Na^+^ and K^+^ elicited Cl^−^/HCO_3_^−^ exchange in Slc4a8-(**A**) and Slc4a10-expressing cells (**C**) loaded with BCECF. Cells were perfused with NMDG-containing high Cl^−^ solution (Solution (**C**), Table 1) followed by NMDG-containing low Cl^−^ solution (Solution (**D**), Table 1) and finally switched to Na^+^ (Solution B, black circles; *n* = 10 for Slc4a8 and Slc4a10) or K^+^-containing solution (Solution G, open circles; *n* = 9 for Slc4a8 and *n* = 10 for Slc4a10). Fluxes obtained from non-electroporated cells are shown as red dashed lines and are the same in A and C (*n* = 16). (**B**,**D**) Slc4a8 (**B**) and Slc4a10 (**D**) activities were measured as the alkalinization rate in response to a reduction in external Cl^−^ concentration using Na^+^- and K^+^-containing solutions (change from solution A to solutions 5, 25, 50, 100, 125 and 150 mM [see Table 2 for ion composition of the Na^+^- and K^+^-containing solutions]). The normalized activities at different Na^+^ (black circles) and K^+^ concentrations (open circles) were plotted against the cation concentration and then fitted to a Hill function (shown as black lines; see Methods for the equation used). Values correspond to the average ± SEM.

**Figure 4 ijms-25-04575-f004:**
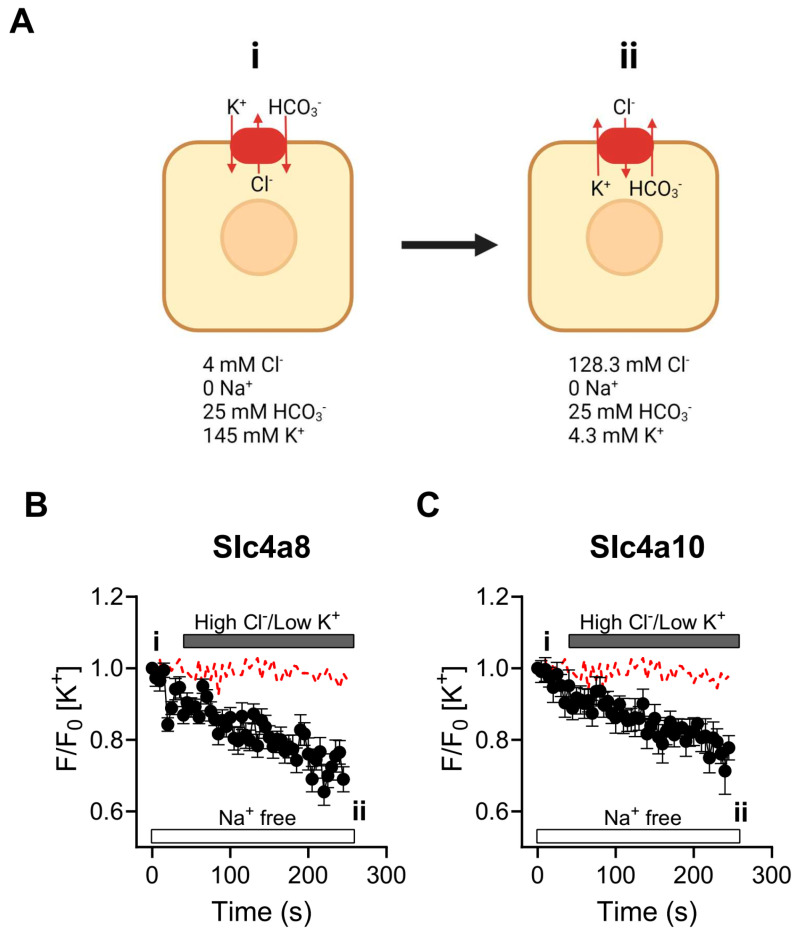
Slc4a8− and Slc4a10−dependent K^+^- fluxes. (**A**) Experimental design used to assess K^+^ transport by Slc4a8 and Slc4a10. (**B**,**C**) Slc4a8- ((**B**), black circles, *n* = 7) and Slc4a10-expressing cells (**C**, black circles, *n* = 9) loaded with PBFI-AM were depleted of Cl^−^ by incubating the cells with low Cl^−^ solution and high K^+^ (solution G, Table 1) and then Cl^−^ uptake (and K^+^ efflux) was induced by switching to an external solution containing high Cl^−^ and low K^+^ (solution C, Table 1). Red dashed lines in (**B**,**C**) show the activity displayed by non-transfected cells (*n* = 16). The experiments shown in (**B**,**C**) were performed under Na^+^-free conditions to prevent Na^+^ contamination of the PBFI signal. Results are presented as the mean ± SEM.

**Figure 5 ijms-25-04575-f005:**
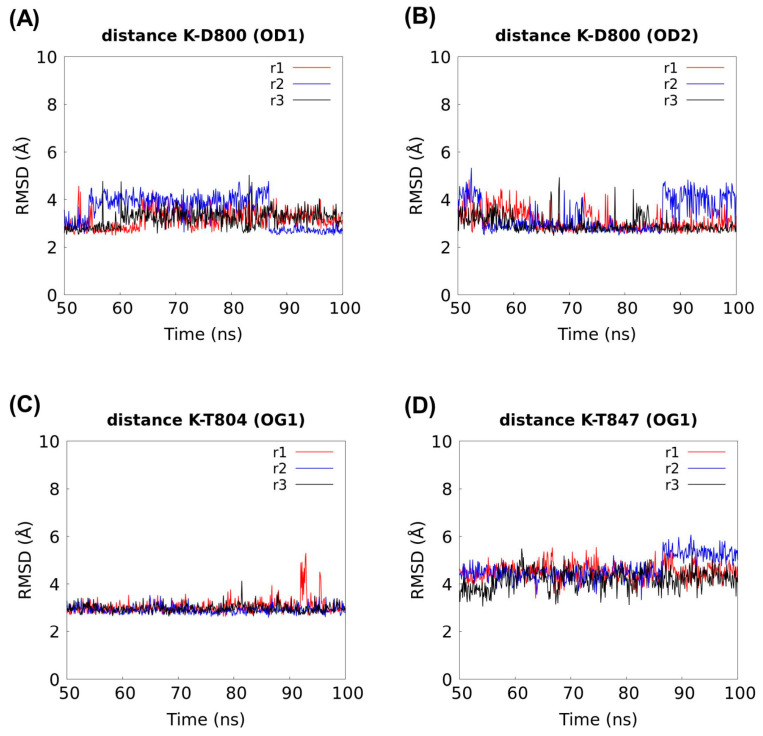
K^+^ occupancy at the S1 site in the cryo-EM rSlc4a8 structure. Distances between K^+^ and the residues from S1 Na^+^-coordination site were computed within a stable range along three 100 ns MD simulations (referred to as r1, r2, and r3). Residues for distance calculation include D800 and its carbonyl side chain atoms OD1 and OD2 (**A**,**B**), and the respective hydroxyl OG1 side chain atoms from T804 (**C**), and T847 (**D**).

**Figure 6 ijms-25-04575-f006:**
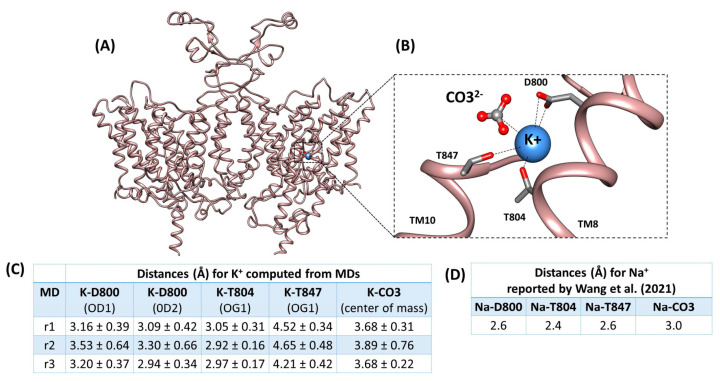
Residues at the S1 site involved in K^+^ coordination. Ribbon representation of (**A**) MD equilibrated structure of rat Slc4a8 where K^+^ replaces Na^+^ at S1 site from 100 ns MD running and (**B**) K^+^ interactions at S1 site with the residues originally described coordinating Na^+^ in [17]. The residue side chains are shown as sticks, CO_3_^2−^ is shown as ball and sticks and, K^+^ is shown as a blue sphere. The transmembrane helices, TM8 and TM10, contributing to ion coordination are labeled. (**C**). The average distances between K^+^ and residues D800, T804, T847 and the CO_3_^2−^ ion computed from MD simulations are indicated (referred as r1, r2, and r3). The K^+^-residue distances are measured to side chain atoms, in particular, the oxygen atoms (OD1 and OD2) from the carbonyl group of D800 and the oxygen (OG1) from hydroxyls of T804 and T847. The distance between K^+^ and CO_3_^2−^ was computed between their center of mass. (**D**) Na^+^ contacting distance with residues D800, T804, T847, and the CO_3_^2−^ ion was taken from Wang et al. [17].

**Figure 7 ijms-25-04575-f007:**
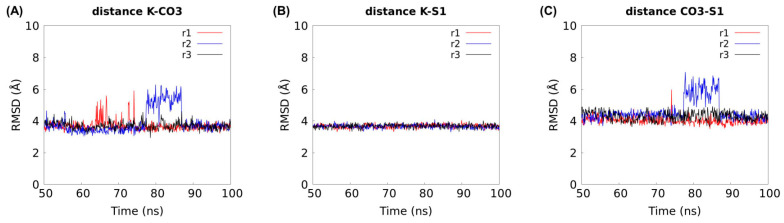
Interaction between K^+^ and CO_3_^2−^ at the S1 site. Distance between (**A**) K^+^ and center of mass of CO_3_^2−^, (**B**) K^+^ to S1 site, and (**C**) CO_3_^2−^ to S1 site along three 100 ns MD simulations (referred to as r1, r2, and r3). The S1-ion distance is defined as the distance between each ion and the Cα atom of residue A846, similar to the definition used by Wang et al. [17]. The residue A486 belongs to the S1 site.

**Table 1 ijms-25-04575-t001:** Solutions used in this study.

Component	Solution Name						
(mM)	A	B	C	D	E	F	G
Na^+^	145	145			145	145	
K^+^	4.3	4.3	4.3	4.3	4.3	4.3	145
NMDG			120	120			
Ca^2+^	1	1	1	1	1	1	1
Mg^2+^	1	1	1	1	1	1	1
Choline			25	25			
Cl^−^	128.3	4	128.3	4	153.3	4	4
HCO_3_^−^	25	25	25	25			25
Gluconate		124.3				149.3	120
Glutamate				124.3			
Glucose	5	5	5	5	5	5	5
HEPES	10	10	10	10	10	10	10
pH	7.4	7.4	7.4	7.4	7.4	7.4	7.4

HCO_3_^−^-containing solutions were gassed with 5% CO_2_/95% O_2_. HCO_3_^−^-free solutions were continuously gassed with 100% O_2_.

**Table 2 ijms-25-04575-t002:** Solutions used in cation selectivity experiments.

Solution (mM)	Na^+^ 5	Na^+^ 25	Na^+^ 50	Na^+^ 100	Na^+^ 125	Na^+^ 150	K^+^ 5	K^+^ 25	K^+^ 50	K^+^ 100	K^+^ 125	K^+^ 150
Na^+^	5	25	50	100	125	150						
K^+^							5	25	50	100	125	150
NMDG	120	125	100	50	25		120	125	100	50	25	
Ca^2+^	1	1	1	1	1	1	1	1	1	1	1	1
Mg^2+^	1	1	1	1	1	1	1	1	1	1	1	1
Choline	25						25					
Cl^−^	4	4	4	4	4	4	4	4	4	4	4	4
HCO_3_^−^	25	25	25	25	25	25	25	25	25	25	25	25
Glutamate	125	125	125	125	125	125	125	125	125	125	125	125
Glucose	5	5	5	5	5	5	5	5	5	5	5	5
HEPES	10	10	10	10	10	10	10	10	10	10	10	10
pH	7.4	7.4	7.4	7.4	7.4	7.4	7.4	7.4	7.4	7.4	7.4	7.4

HCO_3_^−^-containing solutions were gassed with 5% CO_2_/95% O_2_. HCO_3_^−^-free solutions were continuously gassed with 100% O_2_.

## Data Availability

The original contributions presented in the study are included in the article/Appendix A, further inquiries can be directed to the corresponding author.

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
