# Peer review of "K^+^-Driven Cl^−^/HCO_3_^−^ Exchange Mediated by Slc4a8 and Slc4a10"

_ijms, 2024, doi:10.3390/ijms25084575_

Round 1
Reviewer 1 Report
Comments and Suggestions for Authors
Slc4a genes encode various transporters, including Na+-HCO3- cotransporters, Cl-/HCO3- exchangers, or Na+-driven Cl-/HCO3- exchangers. Previous research has revealed that Slc4a9 (Ae4) functions as a Cl-/HCO3- exchanger, which Na+ or K+ can drive. In the present study, the authors show that either Na+ or K+ drives Cl-/HCO3- exchanger activity in cells overexpressing Slc4a8 or Slc4a. Further, they demonstrated that Slc4a and Slc4a also mediate Cl- and HCO3- -dependent K+ transport.
Overall, this manuscript is well-designed, and the conclusions are sound. However, essential control experiments still need to be included. It must be shown that mock control cells do not show such a change of the fluorescence shown in Figs, which indicates that this reaction is due to a consequence of ion transport through the expressed slc4a. In addition, the percentage of cells that express recombinant transporters should be stated in the text.
Author Response
We express gratitude to the reviewer for their valuable feedback. In the revised manuscript, we have incorporated traces from control cells into all figures featuring functional experiments. Additionally, we have included the percentage of cells expressing recombinant Slc4a8 and Slc4a10 transporters in the Methods section.
Reviewer 2 Report
Comments and Suggestions for Authors
This manuscript addresses just one point in enough detail to demonstrate what has happened in the course of the transport that it describes, but does so carefully and with sufficient data to make that point, concerning the role of K+ in what is otherwise an anion transporter. There is not enough in the manuscript to really define a full mechanism, and the MD calculation really cannot do this in a convincing way (here it would be possible to include a longer review, but at this time it is not needed). However, the experimental evidence does appear to be sufficient to demonstrate that there is a role for K+ in the exchange mediated by this transporter; a thorough discussion of the mechanism would be more interesting, but can be left for later work to determine the details. Clarifying the point that is addressed in detail is of sufficient interest to merit publication.
The English is good enough, and the paper can be published as is.
Author Response
We are grateful to the reviewer for their insightful feedback.